# Aloft Transport of Haze Aerosols to Xuzhou, Eastern China: Optical Properties, Sources, Type, and Components

Kai Qin [1], Qin He [1], Yishu Zhang [1], Jason Blake Cohen [1], Pravash Tiwari [1] and Simone Lolli [2],*

1 School of Environment and Spatial Informatics, China University of Mining and Technology, Xuzhou 221116, China; qinkai@cumt.edu.cn (K.Q.); heqin@cumt.edu.cn (Q.H.); zhangyishu@cumt.edu.cn (Y.Z.); wjjs0011@cumt.edu.cn (J.B.C.); tiwarip@cumt.edu.cn (P.T.)
2 CNR-IMAA, Consiglio Nazionale delle Ricerche, Contrada S. Loja snc, 85050 Tito, PZ, Italy
* Correspondence: simone.lolli@imaa.cnr.it

**Abstract:** Rapid industrialization and urbanization have caused frequent haze pollution episodes during winter in eastern China. Considering that the vertical profile of the aerosol properties changes significantly with altitude, investigating aerosol aloft information via satellite remote sensing is essential for studying regional transport, climate radiative effects, and air quality. Through a synergic approach between lidar, the AErosol RObotic NETwork sunphotometer observations, and WRF-Chem simulations, several transboundary aloft transport events of haze aerosols to Xuzhou, eastern China, are investigated in terms of source, type, and composition and the impact on optical properties. Upper-air aerosol layers are short-lived tiny particles that increase the total aerosol optical depth (AOD). The aloft aerosols not only play a critical role during the haze event, enhancing the scattering of aerosol particles significantly but also cause a rise in the AOD and the Ångström exponent (AE), which increases the proportion of fine particles, exacerbating the pollution level near the surface. Based on the model simulation results, our study highlights that the transported aloft aerosols lead to the rapid formation of secondary inorganic substances, such as secondary sulfates, nitrates, and ammonium salts, which strongly contribute to haze event formation. Moreover, the results provide evidence that the haze frequency events associated with polluted dust outbreaks were higher for 2014–2015 winter. A closer analysis shows that the advected dust layers over Xuzhou originated from Inner Mongolia and the Xinjiang Uygur Autonomous Region. The study of the occurrence frequency, height, thickness, and optical properties of aloft anthropogenic haze in China will further deepen our understanding and provide a strong basis to assess aerosol impact on transport and the Earth–atmosphere radiative balance.

**Keywords:** transboundary aloft transport; aerosol optical properties; lidar; CALIPSO; WRF-Chem

## 1. Introduction

Many studies have shown that the vertical profile of the aerosol properties may vary significantly with respect to latitude/longitude and seasons. Assessing aloft aerosol microphysical and optical properties is important to study the regional transport and climate radiative effects [1,2]. Moreover, upper-air aerosol layers introduce a bias in the assessment of air quality from satellite retrievals because their contributions at different altitudes are summed over the entire atmospheric column, yielding a higher aerosol optical depth (AOD) value than would otherwise be used as a proxy to evaluate the surface $PM_{2.5}$ concentration [3]. In the last decade, the scientific community has made efforts to quantitatively assess the aerosol properties both for natural and anthropogenic emissions. In the next paragraph, we review the principal results highlighted by those studies.

Ref. [4] reported the presence of higher-altitude aerosol layers during March/April 2005 and 2006 over Visakhapatnam, on the east coast of peninsular India, observed by a micropulse Light Detection and Ranging (lidar). Ref. [5] investigated the physical and optical properties of the transboundary biomass burning smoke over Singapore by analyzing

Aerosol Robotic NETwork (AERONET) sunphotometer and micropulse lidar measurements. The lidar, which rapidly developed after carbon dioxide laser invention [6], is widely used for research in atmospheric studies because it has a high temporal and spatial resolution. Lidars are currently used to retrieve the atmospheric profiles of the geometrical and optical properties of clouds, aerosols, and precipitation [7]. Ref. [8] observed multiple wildfire smoke layers reaching western Canada from Boreal Asia, by lidar measurements, that led to a substantial increase in particulate concentrations near the surface. Ref. [9] detected strong pollutant outflows from the Mexico City metropolitan area during winter using Cloud-Aerosol Lidar Infrared Pathfinder Satellite (CALIPSO) lidar measurements (details are described in Section 2.2). The outflows often flowed as far north as the Texas coast. Ref. [10] evaluated the impacts of aloft aerosol plume and aerosol type on the correlation of AOD–particulate matter (PM). During the Two-Column Aerosol Project (TCAP) from June 2012 to June 2013, ref. [11] found aloft aerosol layers in both the Cape Cod and maritime columns using a high-spectral resolution lidar. The layers contributed up to 60% of the total AOD. Ref. [12] observed persistent elevated aerosol layers during the monsoon onset period over Kanpur and found that their radiative effect increases the stability of the lower troposphere. Ref. [13] characterized a vast difference in the vertical aerosol loading on a day-to-day basis over Continental Southeast Asia based on whether it was a high-biomass-burning or low-biomass-burning day using CALIPSO data. Ref. [2] showed that aloft-biomass-burning aerosols are advected toward Penang Island, Malaysia, during monsoon seasons, exacerbating haze episodes. Ref. [14] demonstrated that vertically lofted aerosol layer heights are connected with the emissions profiles of co-emitted species, in addition to the fire radiative power (FRP) and the fuel type, as measured by MISR stereoheights. Ref. [15] further characterized that smoke emitted at a height above 850 mb from Continental Southeast Asia was caught in a different dynamical pattern and advected all the way to Singapore, Malaysia, and Sumatra.

Rapid industrialization and urbanization have led to serious air pollution problems in China, leading to a reduction in lifespan, reduced solar energy production [16], and disruptions in transportation, among other issues. In January 2013, a heavy haze with record-breaking $PM_{2.5}$ (particulate matter with diameters smaller than 2.5 μm in aerodynamics) concentrations hit most parts of central-eastern China and attracted a lot of attention from the world over [17–19]. Thereafter, during fall and winter seasons, persistent regional haze pollution have frequently occurred in central-eastern China because of stable synoptic conditions with weak surface wind and strong vertical temperature inversion, which are unfavorable for pollutant dispersion [20–22]. The fine mode particles (i.e., $PM_{2.5}$) responsible for the Chinese winter haze pollution episodes are mostly caused by anthropogenic emissions, including those from industries, vehicles, and secondary aerosol formation from precursors ($NO_x$, $SO_2$, $NH_3$, and VOCs) [23]. The accumulated anthropogenic aerosols present near the surface during haze pollution conditions could be lifted upward by vertical turbulent mixing or specific dynamical conditions, such as passing fronts [24], forming elevated haze layers that subsequently advect to distant regions [25]. There has been a significant improvement in China's air quality since 2013 because of the implementation of strict emission reduction measures as a result of the clean air action program [26–28]. However, the haze episodes frequently continue to impact the megacities [29,30]. A study conducted by [31] showed that the regional haze in winter formed in the North China Plain can be advected to north-east China in ~1−3 days. Previous studies have also revealed that during winter, there is an alternate cycle of haze pollution events over the North China Plain and the Yangtze River delta (YRD), where the YRD episodes are linked to long-range transport driven by cold fronts from the North China Plain [32,33]. This trans-regional interlinked haze transport indicates that megacities, such as Shijiazhuang, Zhengzhou, Jinan, and Xuzhou, located across these two regions are critically impacted by this phenomenon and should be carefully studied.

Xuzhou City, Jiangsu Province, China, is located at the junction of Jiangsu, Shandong, Henan, and Anhui Provinces (Figure 1). Xuzhou, a heavily industrialized metropolitan area,

hosts a large number of local emission sources, accounted for mostly by coal mining and combustion, coal-fired plant construction projects, and industrial smoke [34–38]. Moreover, Xuzhou is affected by road transportation pollution, being the metro region located in the middle of two highly polluted areas, the Beijing–Tianjin–Hebei region (Jing-Jin-Ji) and the Yangtze River Delta (YRD).

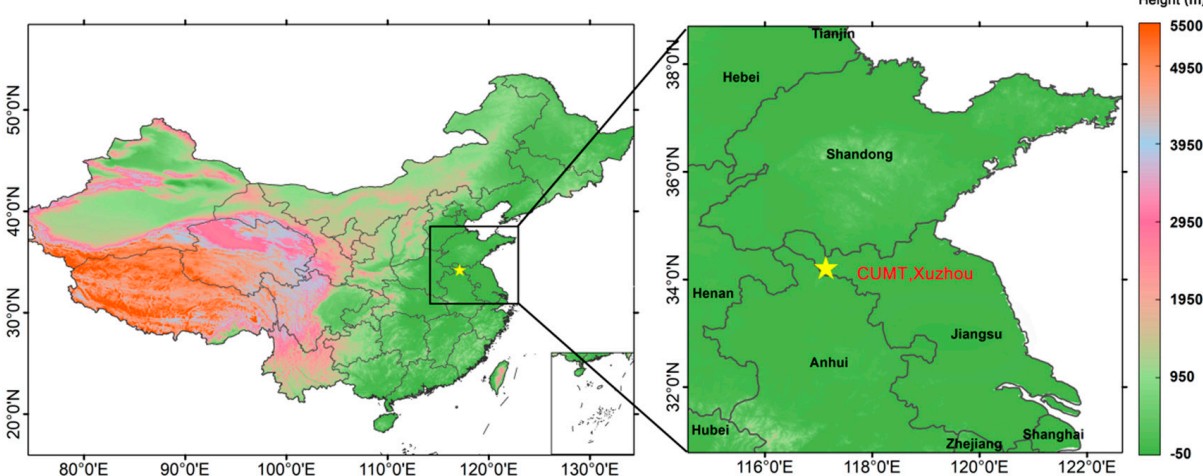

**Figure 1.** Elevation and location map of China University of Mining and Technology, Xuzhou, in China.

The authors analyze and quantitatively assess a trans-regional transport episodes of aloft aerosols advected into Xuzhou [39]. To deepen our understanding of an aloft anthropogenic haze, this study aims to investigate multiple events of transported aerosols to Xuzhou in terms of sources, types, and components and to assess their impacts on optical properties. This rest of the paper is organized as follows: Section 2 describes the observations, the corresponding retrieval methods, and the model setting. Section 3 presents the result analysis from ground- and satellite-based observations and model simulations. Section 4 concludes with the main findings.

## 2. Observations and Model

### 2.1. Ground-Based Observations

The ground-based observational site is located in suburban Xuzhou, eastern China (see Figure 1). The region surrounding the site is rural and consists mainly of hills and lakes, without relevant sources of anthropogenic aerosols. From December 2014 to January 2015, an MPLlidar (model MiniMPL) from Sigma Space Corporation (now Droplet Measurement Technologies) was deployed on the roof of the building of the School of Environment Science and Spatial Informatics, China University of Mining and Technology (34.22°N, 117.14°E, 60 m). A Cimel CE-318 sunphotometer is also permanently deployed on the roof of the building since June 2013, entitled Xuzhou-CUMT, under the framework of the Aerosol Robotic NETwork (AERONET, [40]).

The MiniMPL is capable of vertically resolving geometrical and optical aerosol and cloud properties. The 532 nm laser source is pulsed with a high pulse-repetition frequency (2500 Hz) with low energy laser (3–4 μJ) to meet eye-safe requirements for atmospheric measurements. The MiniMPL sounds the atmosphere from about 120 m (below which, the instrument is "blind" because of the overlap) up to 15 km with a vertical resolution of 30 m. The backscattered signal is corrected for different background noises (including sunlight) and instrumental effects (including dead time, afterpulse, and overlap) to obtain the normalized relative backscatter (NRB). More details on the calibration process can be found in [41,42]. The MiniMPL is equipped with parallel ($\beta_{\parallel}$) and cross-polarized ($\beta_{\perp}$) channels, respectively [43]. The backscattered power by the atmosphere is split into co- and

cross-polar channels. The volume depolarization ratio (VDR) is obtained taking the ratio between the cross- and co-polar backscattering coefficients, as showed in Equation (1):

$$VDR = \frac{\beta_\perp}{\beta_\parallel} \tag{1}$$

The extinction coefficient profile from the MPL is retrieved using the [44] inversion algorithm. The methodology shows large uncertainties because it requires solving a transcendent equation with two unknowns (the extinction coefficient and the backscattering coefficient). To solve this, a conjecture, the lidar ratio (LR) value, which is the ratio between the extinction and backscatter coefficients, along the vertical profile must be assumed. Furthermore, to reduce the uncertainty, it is possible to retrieve the LR from the co-located sunphotometer measurements and constrain the lidar equation through a recursive algorithm [45]. Notably, the lidar ratios can vary in the vertical column, which is not considered in this study. Errors caused by the algorithm and the assumption of a constant lidar ratio are given by [45].

The automatic CE-318 sunphotometer scans direct Sun irradiance four times in an hour at eight spectral channels (340, 380, 440, 500, 675, 870, 1020, and 1640 nm) with a 1.2° full field of view. The measurements of direct Sun irradiance for each band can be employed to calculate the AOD. Ref. [46] estimated that the total uncertainty associated with the AOD for a newly calibrated field instrument is approximately 0.01–0.02. AE (Ångström exponent) is estimated at two wavelengths, 870 nm and 440 nm, which is commonly used to describe the wavelength dependence of AOD and to provide some basic information on aerosol size distribution. A spectral deconvolution algorithm (SDA) was developed to infer the component fine and coarse mode optical depths at 500 nm from the spectral total extinction AOD data based on two fundamental assumptions [47]. The first is that the aerosol particle size distribution is effectively bimodal. The second hypothesis is that the coarse-mode Ångström exponent and its spectral variation are both approximately neutral. There are three levels of AERONET data (http://aeronet.gsfc.nasa.gov; last accessed: 20 March 2022). The data sets used here are Version 3 (V3) taken from the Level 1.5 (cloud screened) product.

## 2.2. Satellite Observations

The Cloud-Aerosol Lidar Infrared Pathfinder Satellite (CALIPSO) launched in 2006 is part of the satellite A-Train constellation. Cloud-Aerosol Lidar with Orthogonal Polarization (CALIOP) loaded on CALIPSO is a dual-wavelength (532 and 1064 nm) elastic backscatter lidar with depolarization channel capability at 532 nm [48]. In this study, 532 nm total attenuated backscatter coefficient (TABC) from CALIPSO Level 1b products and vertical feature masks (VFMs) and aerosol subtype (AS) from CALIPSO Level 2 Version 4.20 products were used, which are available from the NASA Langley Research Center (https://www-calipso.larc.nasa.gov/; last accessed: 20 March 2022). Among them, the 532 nm TABC image is the sum of 532 nm parallel and perpendicular return signals and has been color coded such that blues correspond to molecular scattering and weak aerosol scattering and aerosols generally show up as yellow/red/orange. Stronger cloud signals are plotted in gray scales, while weaker cloud returns are similar in strength to strong aerosol returns and coded in yellows and reds. The AS image shows the vertical and horizontal distribution of different aerosol subtypes, including clean marine, dust, polluted continental, clean continental, polluted dust, and smoke [49]. VFM images [50] show the vertical and horizontal distribution of cloud and aerosol layers in different feature types, such as clear air, cloud, aerosol, stratospheric feature, surface, subsurface, and no signal (totally attenuated).

## 2.3. WRF-Chem Simulations

In this study, version 3.8.1 of the air quality model Weather Research and Forecasting (WRF) with Chemistry (WRF-Chem) is implemented. The WRF model is a mesoscale

non-hydrostatic meteorological model that includes several options for physical parameterizations of planetary boundary layer (PBL), land surface, and cloud processes. WRF-Chem is the WRF model coupled with chemistry that can simultaneously calculate meteorological elements and atmospheric chemical components to achieve "online" coupling. A detailed description of the model is given by [51]. In this study, the simulation covers eastern China, with a central point of 109.4°E and 36°N, including 84 grids in the east–west direction and 92 grids in the north–south direction, with a spatial resolution of 45 km (under the lambert projection). The model accounts for 30 vertical dynamic levels, i.e., the vertical resolution is higher in the lower part of the atmosphere. Meteorological data are driven by NCEP FNL (Final) Operational Global Analysis data at $1° \times 1°$ resolution (https://rda.ucar.edu/datasets/ds083.2/; last accessed on 20 March 2022). The scheme of physical and chemical processes adopted here is listed in Table 1. Anthropogenic emissions are taken from the Multi-resolution Emission Inventory for China (MEIC) database (http://meicmodel.org/, last accessed on 20 March 2022), which provides the complete annual inventory of atmospheric pollutants and greenhouse gases (i.e., $SO_2$, Nox, CO, NMVOC, $NH_3$, $CO_2$, $PM_{2.5}$, $PM_{10}$, BC, and OC) over China in 2012 with a resolution of $0.25°$ [52].

**Table 1.** WRF-Chem Configuration.

| Module | Parameter Scheme |
|---|---|
| Microphysics | WSM 6 |
| Surface layer | Monin–Obukhov |
| Boundary layer | YSU |
| Longwave radiation | Rrtmg |
| Shortwave radiation | Rrtmg |
| Cumulus convection | New Grell |
| Land surface process | Noah |
| Aerosol | MOSAIC (4 bins) |
| Photolysis | Fast-J |
| Biogenic emission | Gunther |
| data | Data |

## 3. Results

### 3.1. Identifying Aloft Haze Plumes Using Ground-Based Lidar Measurements

The ground-based lidar observations were used to detect the aloft-haze-plume events. Aloft haze layers are visually identified as being aloft-haze-plume events if the layer lies more than 300 m above the planetary boundary layer (PBL) height for more than 3 h [53]. The lidar cross-polarization channel helps to identify spherical from non-spherical aerosols (e.g., dust and ash) or thick clouds that cause multiple scattering [54] and ice clouds [55]. Hence, aloft haze layers consisting mainly of fine mode and spherical aerosols, which can be detected by analyzing co-polarized and cross-polarization channels' NRB returns.

The MiniMPL NRB at 532 nm is shown in Figure 2 (the upper and lower panels are cross-polarized (a,c) and co-polarized returns (b,d), respectively, exhibiting the vertical structure and the temporal variability of aerosols). Four aloft-haze-plume events were observed to occur in the framework of this study: from 2 to 4 December 2014, from 7 to 8 December 2014, from 3 to 4 January 2015, and from 7 to 8 January 2015. A common theme in these events is that the aerosol plume is observed to descend into the PBL from the free troposphere. For this reason, the optical thickness within the PBL gradually increases in time as aerosols are entrained into it.

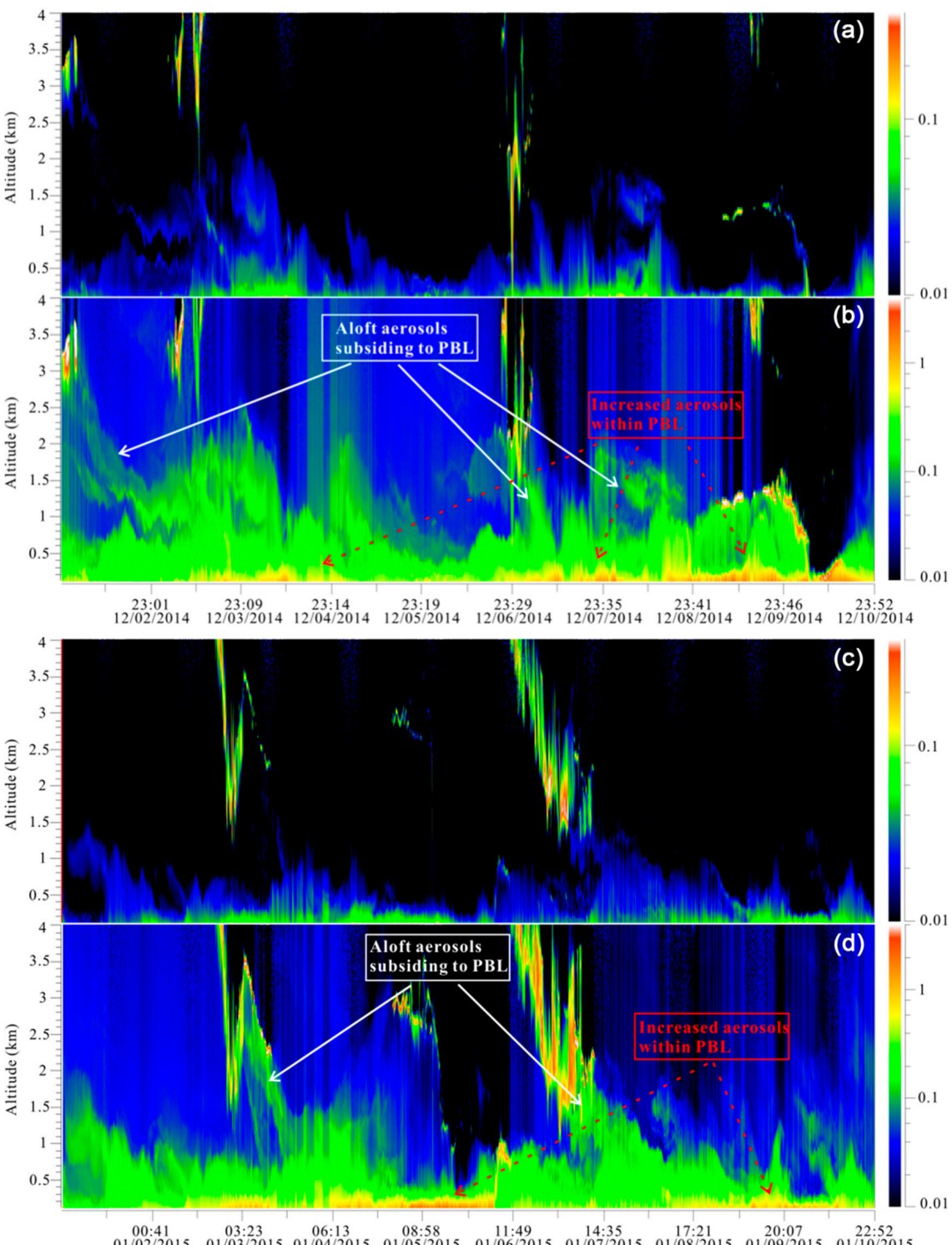

**Figure 2.** Time–height images of MPL NRB (normalized relative backscatter) at 532 nm in cross-polarization (**a**) and co-polarized (**b**) channels for 2–10 December 2014 and in cross-polarization (**c**) and co-polarized (**d**) channels for 1–10 January 2015 (China Standard Time (CST)).

### 3.2. Optical Properties of the Plumes

There were co-located Sunphotometer measurements during the daytime on 2, 4, 7, 8 December 2014 and 3, 4, 7, and 8 January 2015 supporting lidar observations of the aloft aerosol layers. The vertical integration of the extinction coefficient profile yields the columnar AOD. Thus, the lidar-derived extinction coefficient can be retrieved from the

MiniMPL data using the sunphotometer AODs as a constraint so long as the plumes are in the troposphere. The mean vertical profiles of the extinction coefficient in Figure 3a show two aloft layers, one at 1–1.8 km and one at 2.1–2.7 km. The total optical thickness of the two aerosol layers aloft (1–1.8 km and 2.1–2.7 km) is 0.26, accounting for 38% of the total columnar AOD. Figure 3b shows that the depolarization ratio from 120 m up to 3000 m has a consistent mean value of $0.10 \pm 0.01$. This suggests that the aerosols both in the PBL and aloft layers were dominated by spherical particles, consistent with urban sources. Using the sunphotometer measurements from 1 December 2014 to 10 January 2015, a comparison of the days showing aerosol layers aloft against those days without aerosols aloft, in terms of mean values and standard deviations of AOD at 500 nm, AE, and FMF (fine mode fraction), is made. All three parameters are observed to be larger when an aloft haze plume is present, suggesting high aerosol loadings and smaller particles, again consistent with urban sources [56].

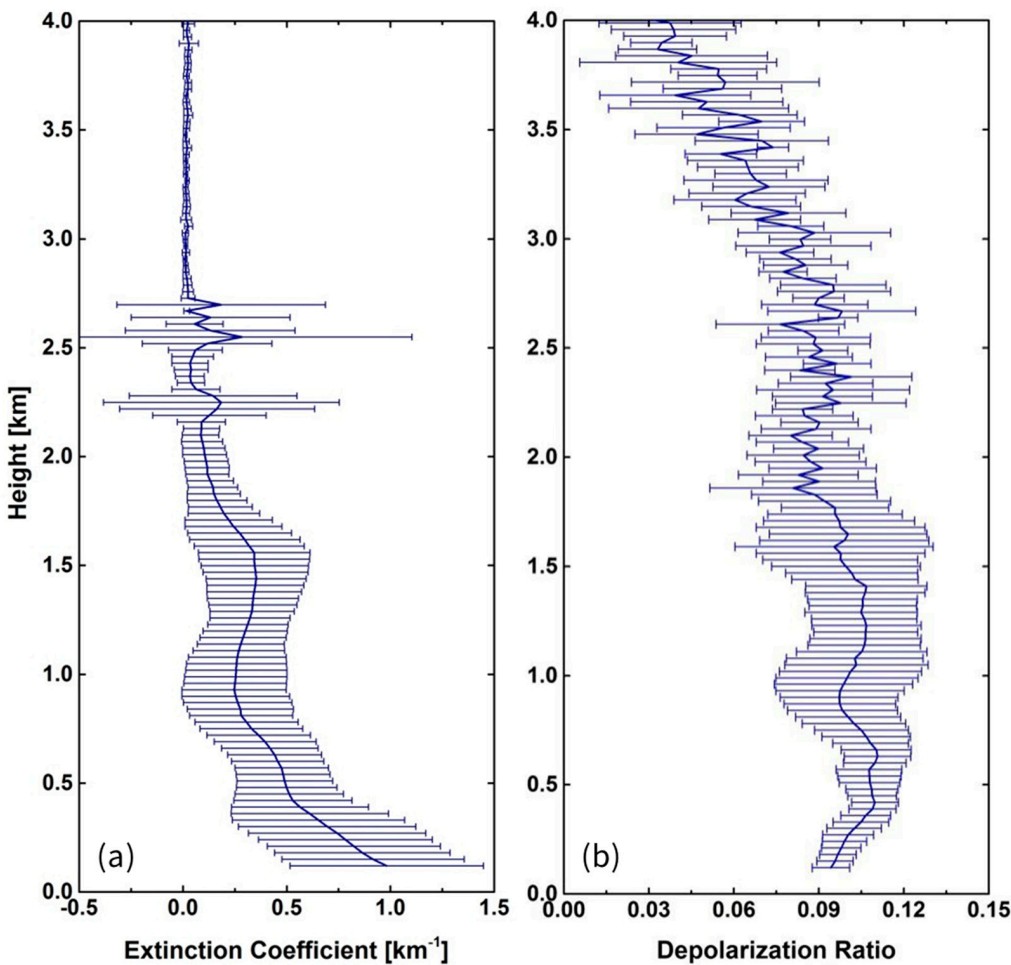

**Figure 3.** The mean vertical profiles of AOD-constrained extinction coefficient (**a**) and depolarization ratio (**b**) during the daytime on 2, 4, 7, and 8 December 2014 and 3, 4, 7, and 8 January 2015.

Figure 4 shows the AE (440–870 nm) and the AOD (440, 675, 870, and 1020 nm) (a,b), the absorption aerosol optical depth (AAOD) (at 440, 675, 870, and 1020 nm) (c,d), and the SSA (440, 675, 870, and 1020 nm) (e,f) during the aloft-haze-plume events. The Ångström exponent is often used as a qualitative indicator of aerosol particle size [57], with values less than 1 indicating large particles, such as sea salt and dust, and values greater than 2 indicating small particles associated with combustion by-products [46,58,59].

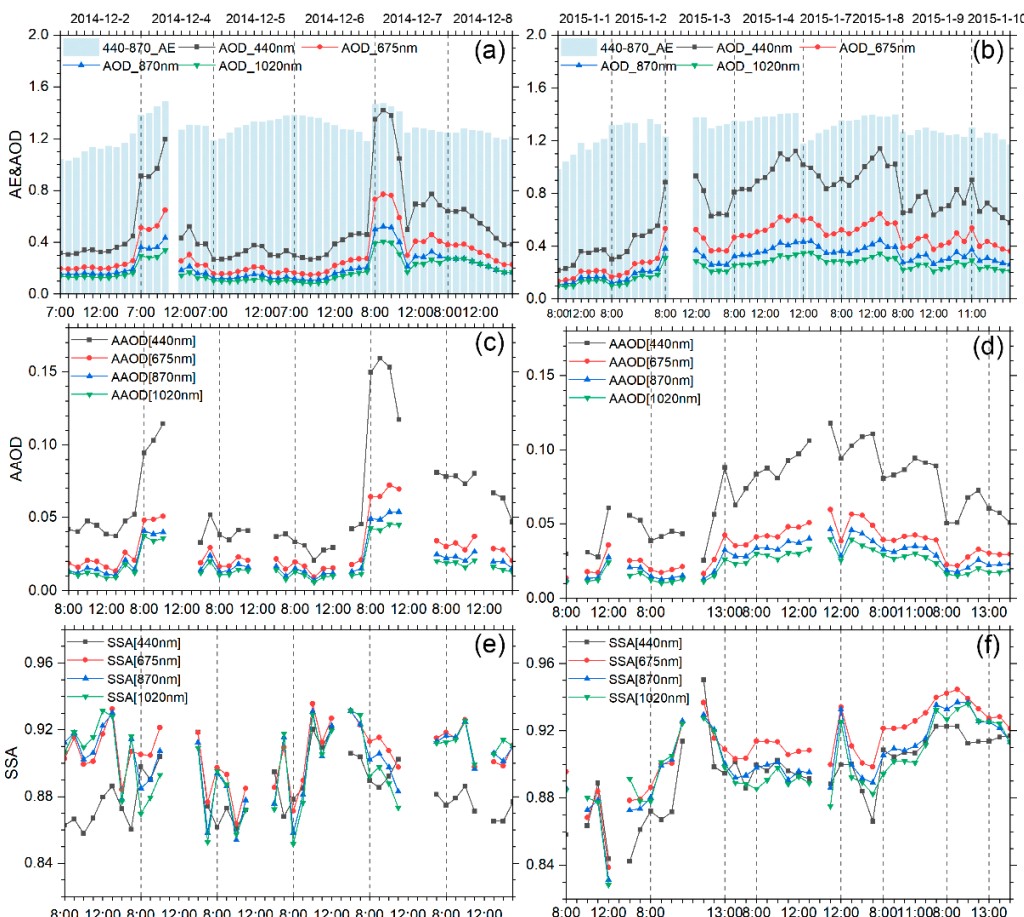

**Figure 4.** Ångström exponent at 440–870 nm (AE440–870 nm) and aerosol optical depth at 440 nm, 675 nm, 870 nm, and 1020 nm (AOD440 nm, AOD670 nm, AOD870 nm, and AOD1020 nm) (**a**,**b**), aerosol absorption optical depth at 440 nm, 675 nm, 870 nm, and 1020 nm (AAOD440 nm, AAOD670 nm, AAOD870 nm, and AAOD1020 nm) (**c**,**d**), and single scattering albedo at 440 nm, 675 nm, 870 nm, and 1020 nm (SSA440 nm, SSA675 nm, SSA870 nm, and SSA1020 nm) (**e**,**f**) during 02 to 08 December 2014 (**a**,**c**,**e**) and 01 to 10 January 2015 (**b**,**d**,**f**) (China Standard Time (CST)).

During the two cases, the values of the AE when there is an aloft layer present and without an aloft layer present were found to be quite different (1.17 ± 0.29 and 1.30 ± 0.12; Table 2), indicating that the proportion of fine particles increased when there were aerosols aloft. Absorbing aerosols, produced by biomass and other combustion processes as well as dust outbreaks [60,61], mainly include black carbon (BC) and mineral dust. Single scattering albedo (SSA) is defined as the ratio of the scattering coefficient to the extinction coefficient and gives important information about aerosol scattering and absorption, including the ability to quantify the abundance of absorbing aerosols. In this experiment, starting on 2 December 2014, aerosols at higher altitudes descended into the boundary layer as far as the ground, while both the AOD and the AE rose slightly. The SSA at most wavelengths reached values above 0.9, while the AAOD changed slightly overall. Although the Sunphotometer data were missing on December 3, the AOD, the AE, the AAOD, and the SSA all increased on December 4 with respect to December 2. Considering that the increase in coal burning in winter will also lead to a rise in the AAOD, and further combined with the time–height vertical profile of the MPL (Figure 2), it was observed that slight pollution occurred near the ground from December 3 to 4. Therefore, the haze pollution was likely to be caused by a combination of local emissions and the transboundary advection of aerosols from aloft. Before 10:00 a.m. China Standard Time (CST) on December 4, it was found that the AOD and the AE increased sharply for all wavelengths, and after 10:00 a.m. CST, the

AOD, the AE, and the AAOD all decreased simultaneously, possibly due to the increase in the atmospheric boundary layer and the subsequent horizontal advection of cleaner air from surrounding surface regions. However, the SSA did not decline until 14:00 CST, and its change lagged behind the AOD and the AE. From 12:00 CST on December 6, the SSA in most bands (except 440 nm) was observed to increase with air quality deterioration, while the AAOD was still observed to be low. From December 6 to the night of December 7, the ground-based lidar data show that the aloft aerosol height gradually decreased. After 11:00 CST on December 7, a sharp increase in the SSA was observed over most of the wavelengths, whereas the AAOD showed the opposite trend, suggesting that aerosol sedimentation of absorbing particles greatly increased the scattering of particulate matter.

**Table 2.** Statistics of the mean values and standard deviations of the AOD at 500 nm, the AE, and the FMF during cases with layers and with no layers measured with a sunphotometer from 1 December 2014 to 10 January 2015.

| | AOD | AE | FMF |
|---|---|---|---|
| No upper layer | $0.40 \pm 0.23$ | $1.17 \pm 0.29$ | $0.70 \pm 0.17$ |
| Upper layer | $0.68 \pm 0.26$ | $1.30 \pm 0.12$ | $0.81 \pm 0.09$ |

In the early morning of 2 January 2015, slight pollution occurred near the ground, leading to a continuous rise in the AOD throughout the day on 2 January 2015, which may have been caused by the accumulation of pollutants. On the morning of January 3, clouds mixed with aerosols at a higher altitude, and then the aerosol layer gradually mixed with the planetary boundary layer. It can be seen that the AOD and the AE started to rise at 14:00 CST on January 3 and the AOD of each wavelength reached the maximum at 14:00 CST on January 4, with the highest observed value being 1.12, while the AE reached its maximum value, of 1.41, at 16:00 CST on January 4, suggesting that transboundary aerosol input increased the proportion of fine particles. On January 2, on account of slight pollution near the ground, the SSA increased sharply and reached a maximum before 12:00 am, reaching 0.95, whereas it showed a downward trend after 12:00 a.m. CST. However, the SSA was relatively constant, at around 0.9, on January 3 and 4 because of considerable aerosol presence at higher altitudes. Meanwhile, affected by pollution caused by local emissions, the AAOD increased from 13:00 to 14:00 CST on January 2. The input of external aerosols caused the level of near-surface pollution to increase once more from 14:00 on January 3 to 15:00 CST on January 4, during which the AAOD showed a continuous growth. During this period, the AAOD at the 440 nm wavelength was significantly higher than that at other wavelengths, suggesting that the 440 nm wavelength is more sensitive to the absorbing aerosols and hence that the aerosols were small in size. On 7 January 2015, lidar observations show the presence of high-altitude aerosols over Xuzhou. Owing to the low absorption of near-ground particles, the SSA gradually declined. On January 8, the aloft aerosols mixed into the atmospheric boundary layer on a large scale and the AOD and the AE increased simultaneously from 08:00 to 13:00 CST. The increase in the SSA on January 8 because of the intensification of near-surface pollution reveals that the input of external aerosols significantly enhanced the scattering of aerosol particles. On the contrary, the change in the AAOD was relatively stable in this process, which further reveals that the contribution of external input played a crucial role in this event and that there is the strong possibility of coatings playing an important role in connecting the SSA with the AAOD, consistent with [62].

The aerosol volume size distribution (dV(r)/dln(r)) between 0.05 μm and 15 μm during the December 2–10, 2014 and January 1–10, 2015 time periods was retrieved using the AERONET inversion algorithm (Figure S1 in the Supplementary material). Affected by four haze transport events, the volume concentration of the fine mode increased substantially. During the severe haze period from 4 December through 7 December 2014, the fine particle mode peaks became more prominent and the volume concentration increased significantly,

the largest reaching up to 0.13, with the peak particle size of about 0.15 μm. Likewise, on 3, 4, 8, and 9 January 2015, the volume concentration of the fine mode particle peak was more prominent concerning the coarse mode, up to 0.1, and the peak particle size could reach 0.15 μm.

### 3.3. Compositions, Aerosol Type, and Sources

To obtain an overview of these lidar-observed aloft-haze-plume events and their transport pattern, MODerate resolution Imaging Spectroradiometer (MODIS) and CALIPSO satellite images were collected paired with the analysis of Hybrid Single Particle Lagrangian Integrated Trajectory (HYSPLIT) model running in GIS-based software TrajStat [63–65]. As shown in MODIS true-color images (Figure 5), widespread hazy clouds can be found over eastern China during these events. Since other strong sources of spherical aerosol (e.g., biomass burning) are absent in winter, and based on the other considerations already mentioned in terms of the SSA, the AOD, the FMF, and the AAOD [15], these should be anthropogenic haze pollution.

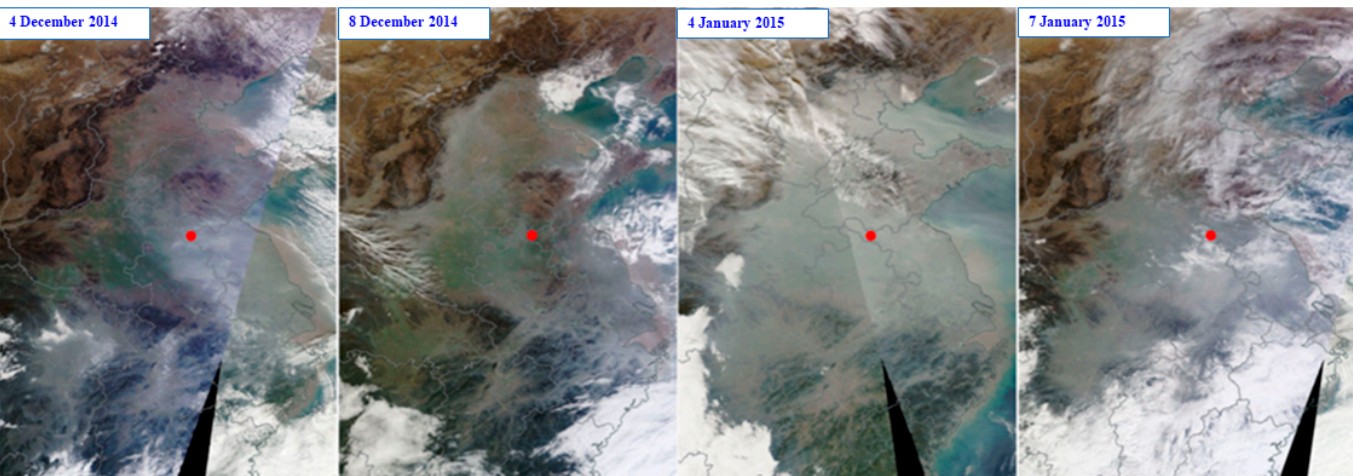

**Figure 5.** AQUA MODIS true-color images during lidar-observed aloft-haze-plume events (China Standard Time (CST)). On the four images, Xuzhou is indicated with a red dot (images sourced from: https://worldview.earthdata.nasa.gov/ (last access 3 February 2022).

To further investigate the components of aloft haze plumes, WRF-Chem was used to simulate the effective days. The simulated PM$_{2.5}$ concentration was compared with the available observational data to evaluate the performance of WRF-Chem during the four haze pollution periods (Figure S2 in the Supplementary material). As shown in Figures 6 and 7, when CALIPSO satellites flew over Xuzhou on 2 and 4 December 2014 and 3 and 5 January 2015, the PM$_{2.5}$ components along the trajectory were extracted to study the distribution characteristics and movement process of PM$_{2.5}$ components in the vertical direction.

At 13:00 on 2 December 2014, PM$_{2.5}$ was mainly found below 2000 m, with a higher concentration mainly below 1000 m. The aloft aerosol layer continuously dropped in height and mixed into the atmospheric boundary layer at 02:00 on 4 December 2014, causing a sharp rise in the near-surface PM$_{2.5}$ concentration, among which the changes in nitrate, organic carbon, and BC are obvious.

At 13:00 on 3 January 2015, the distribution of PM$_{2.5}$ components in the vertical direction was mainly concentrated below 2500 m, while higher concentrations appeared closer to the ground. The concentration of nitrate and ammonium salt in PM$_{2.5}$ was high. On 5 January 2015 at 02:00, PM$_{2.5}$ and its main components were found at higher altitudes (over 2500 m). Nitrate and sulfate concentrations increased significantly at higher altitudes, and rapid formation of secondary inorganic substances was one of the main

causes of haze events [22,23,66], indicating that the high-altitude plume greatly increased the concentration of PM$_{2.5}$ in the air, which further caused the drop in air quality when it settled into the atmospheric boundary layer.

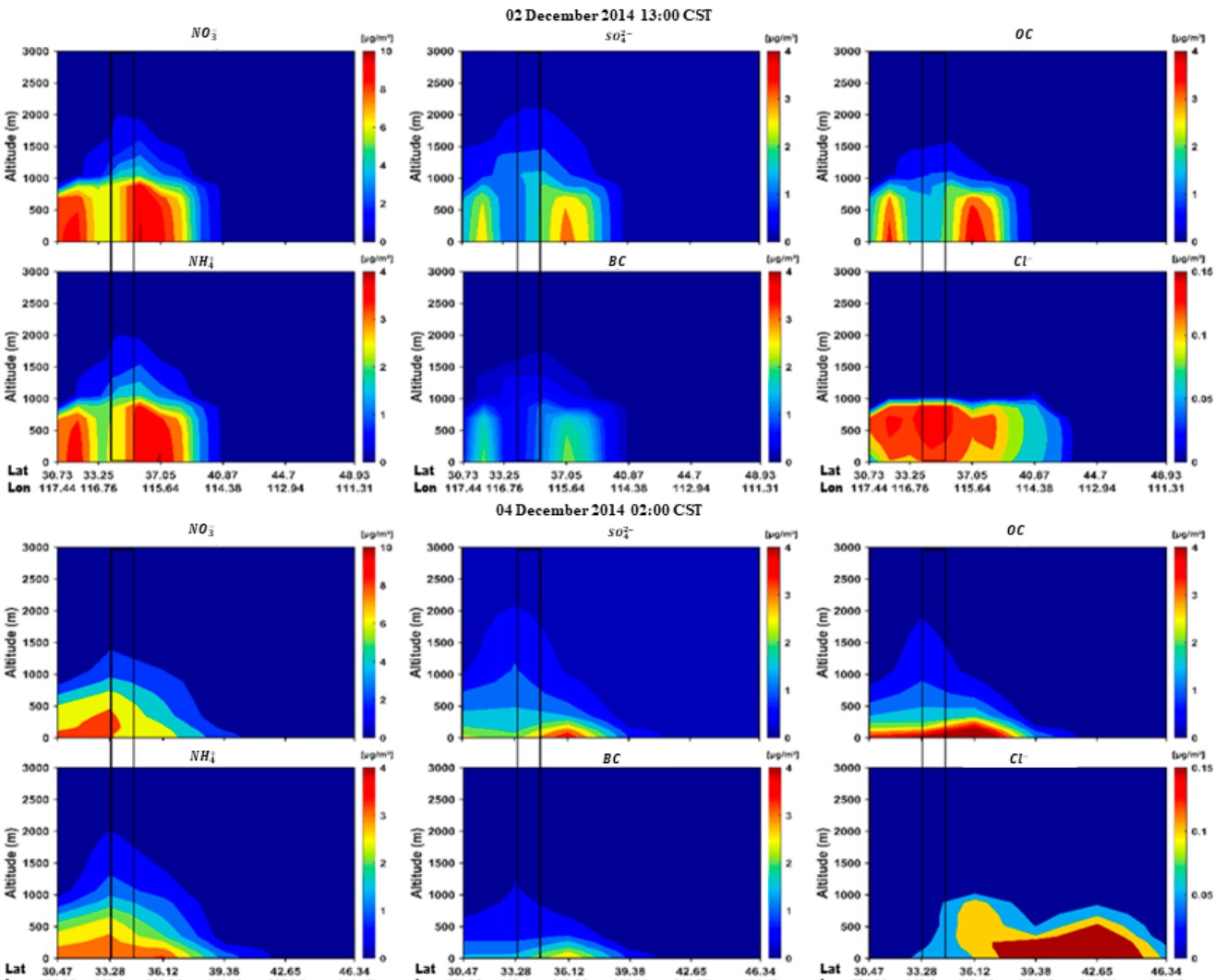

**Figure 6.** Vertical sections of the main components of PM$_{2.5}$ along the trajectory of CALIPSO on 2 December and 4 December 2014 (China Standard Time (CST)). The black box represents the scope of Xuzhou.

Moreover, model simulation results show that during the four haze pollution periods, as the main absorbing aerosol, the concentration of BC in the vertical direction significantly increased at 02:00 on 4 December 2014 and at 02:00 on 5 January 2015 compared with the concentrations at 13:00 on 2 December 2014 and at 13:00 on 3 January 2015.

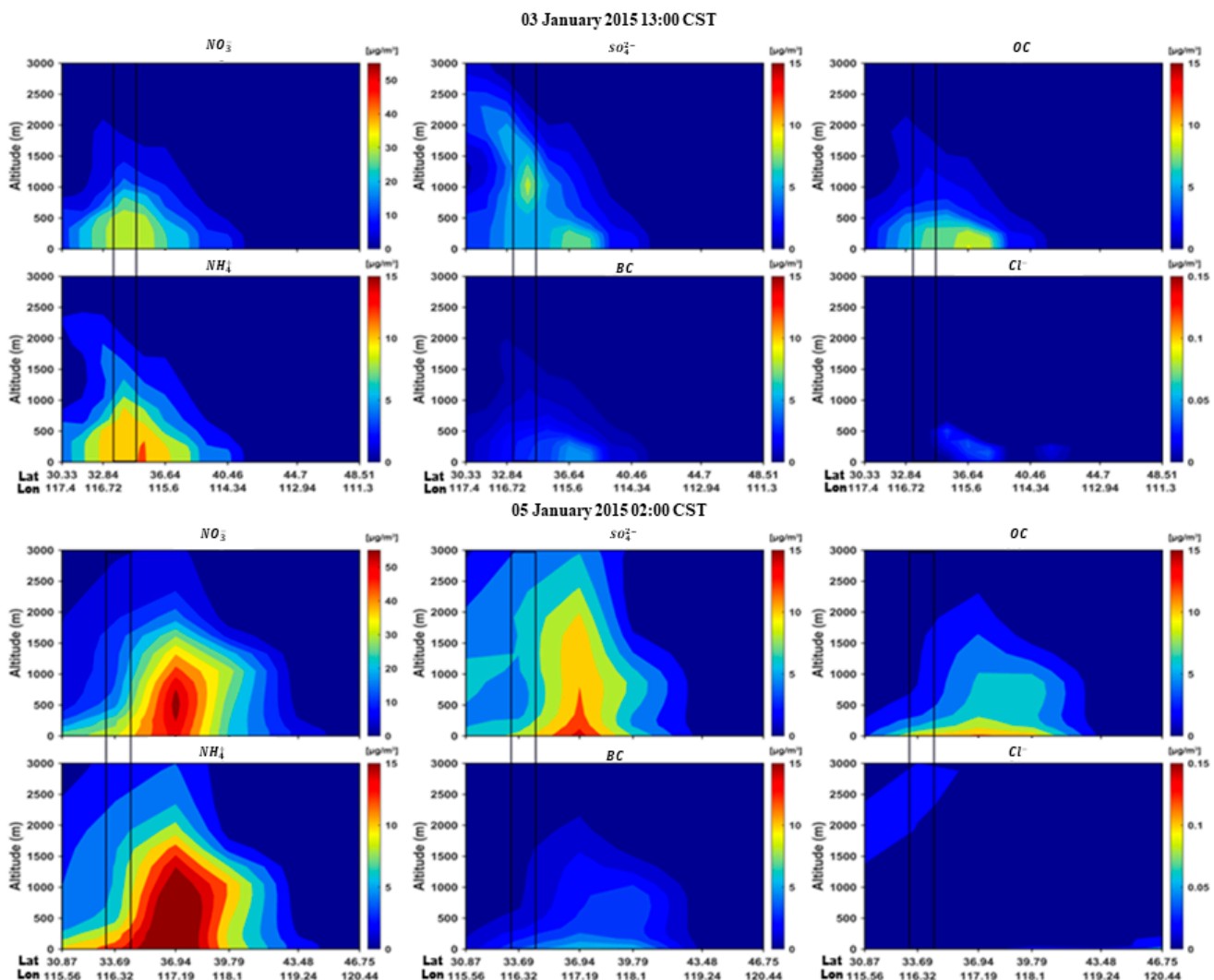

**Figure 7.** Vertical sections of the main components of PM$_{2.5}$ along the trajectory of CALIPSO on 3 January and 5 January 2015 (China Standard Time (CST)). The black box represents the scope of Xuzhou.

To comprehensively study the vertical distribution characteristics of the four aloft-haze-plume events during the analysis period, CALIPSO L2 products were analyzed when the CALIPSO satellites flew over Xuzhou on 2 and 4 December 2014 and 3 January 2015 (the CALIPSO L2 product had no valid signal in or near Xuzhou on January 5). Figure 8 shows the ground tracks of the CALIPSO satellites on 2 and 4 December 2014 and 3 January 2015, with red lines as well as the altitude-orbit cross-section images of 532 nm TABC, VFM, and AS. As can be seen from Figure 8, in addition to the pollutants concentrated near the ground, a pollutant belt was also present at a high altitude, with a height of about 1.5–2 km, which was consistent with the height of the aerosol plume observed by ground-based lidar. At the same time, the AS image clearly reveals the aerosol types at different vertical heights. The aloft polluted areas are classified as aerosols and further identified as polluted dust in the VFM image, which is consistent with a mixture of absorbing and non-absorbing aerosols mostly in the FMF size. It can be clearly seen that the aloft aerosol of the second pollution event on 3 January 2015 has a wider thickness than that of the first haze episode (1.5–3 km). Aerosols at high altitudes were identified as polluted continental and polluted dust in the AS images, which is generally similar but with an even more significant loading in the FMF. These results indicate clearly that the haze events in Xuzhou in winter are of anthropogenic origin, consisting of relatively small particles intermixed with BC.

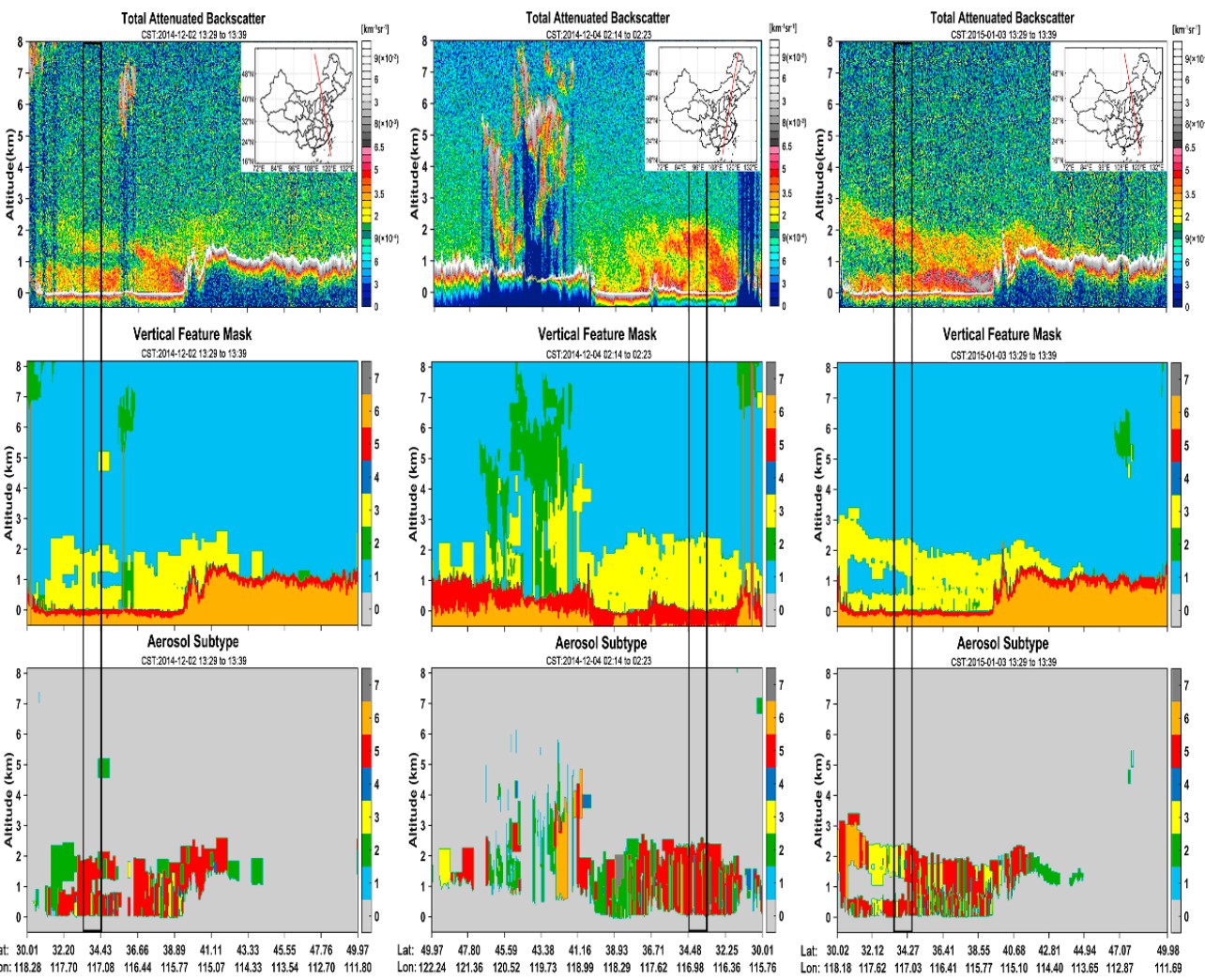

**Figure 8.** CALIPSO 523 nm total attenuation backscattering coefficient, vertical feature masks, and aerosol subtypes on 2 December 2014 (**left**), 4 December 2014 (**middle**), and 3 January 2015 (**right**) orbits and their ground tracks (China Standard Time (CST)). The black box represents the scope of Xuzhou.

The 72 h backward trajectory clustering and weighted potential source contribution function (WPSCF) analysis of the four haze episodes during the analysis period is shown in Figure 9. The back-trajectory ending point is set with respect to the location of the School of Environment Science and Spatial Informatics, China University of Mining and Technology (34.22°N, 117.14°E). The back-trajectory altitude is set at 1500 m. During the period from 2 to 10 December 2014, the average backward trajectories are grouped into three clusters, with 38% of the air parcels (cluster 4) originating from Inner Mongolia, Mongolia, and Xinjiang and transporting eastward via Shanxi, Hebei, and Shandong Province to Xuzhou; 32% of the air parcels (cluster 1) transporting mainly through the Xinjiang Uygur Autonomous Region and entering Xuzhou via Gansu, Shanxi, and Henan Provinces; followed by 16% of the particles originating from Kazakhstan and subsequently following the pathways above (cluster 3). This stands in contrast to the proportion of particles in Xuzhou's surrounding area (cluster 2), which is found to be only about 13%, allowing support for the idea that the majority of the particles during the time of the haze pollution events have a distant source. As can be seen from Figure 9c, the distribution of WPSCF shows that Shanxi, Hebei, Shandong, and Henan are the most likely regions from which the air parcels came. On

January 1–10, it can be seen from Figure 9b that the proportion of air parcels transported into Xuzhou originating from the Xinjiang Uygur Autonomous Region is the largest, reaching 57% (cluster 3), followed by 31% (cluster 2) originating in Sichuan and Chongqing, and subsequently passing through Hubei and Henan Provinces to enter Xuzhou. The proportion of particles from Mongolia and Inner Mongolia was the lowest, at 12% (cluster 1), and were subsequently transported as above, revealing that the particle sources of these aloft transported events are not the same. According to the distribution results of WPSCF (Figure 9d), the most likely sources of PM$_{2.5}$ during 1–10 January 2015 were Shanxi, Hebei, Shandong, and Anhui.

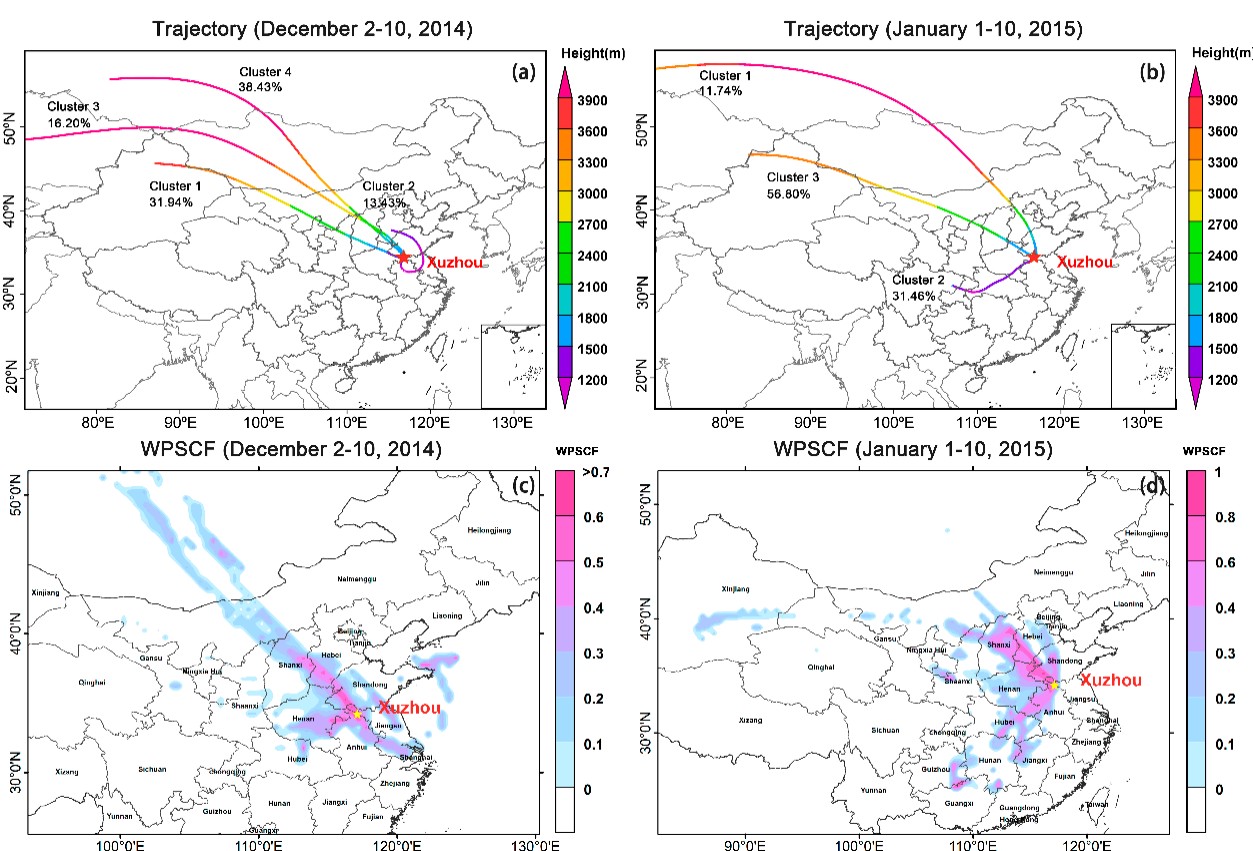

**Figure 9.** Mean 72 h backward trajectories at a 1500 m altitude and WPSCF in Xuzhou during 2–10 December 2014 (**a**,**c**) and 1–10 January 2015 (**b**,**d**).

## 4. Conclusions

Aerosol emissions play a crucial role both because of their radiative effects of the Earth–atmosphere system and in terms of contributing to degraded air quality. Serious haze pollution events occurred in Xuzhou from December 2014 to January 2015. Ground-based lidar and CE-318 sunphotometer observations were used to identify the advected aloft haze plumes and assess their impacts on optical properties. Further analysis was performed using WRF-Chem simulation, CALIPSO satellite images, and basic air parcel backward trajectories to further investigate aerosol composition, type, and source. The main findings and conclusions of the study are summarized below.

Lidar measurements detected four advected aloft-haze-plume events: on 2–4 and 7–8 December 2014 and on 3–4 and 7–8 January 2015. The optical thickness within the planetary boundary layer (PBL) was observed to gradually increase as the aloft aerosols were entrained. The contribution of transported aerosols from distant sources was observed to play a leading role in haze pollution formation during this event. When the aloft haze plume appeared, the aerosol loading and the fine fraction both increased significantly. The

lidar depolarization channel highlights that the advected particles are mainly spherical, consistent with urban sources and providing sufficient time for coatings to appear. Indeed, the average fine mode fraction and depolarization ratios remained at $0.81 \pm 0.09$ and $0.1 \pm 0.01$, respectively. The SSA at all wavelengths was observed to become greater than 0.9 except for the measurements at 440 nm, indicating that the particles are strongly scattering on average, with a small number of significantly small BC cores possibly mixed in. The AOD and the AE were positively correlated and increased simultaneously when the particles settled into the PBL, contributing to the expansion of the proportion of fine particles and exacerbating the pollution level. Besides, during the haze transport cases, the volume concentration of the fine mode was observed to increase dramatically, up to 0.1.

On 2 and 4 December 2014 and 3 January 2015, during CALIPSO satellite overpasses over Xuzhou, CALIPSO L2 products were used to comprehensively analyze the vertical distribution characteristics of aerosols during the haze pollution periods. The aloft polluted regions were classified as aerosols in the AS images and further identified as small absorbing particles in the VFM images, suggesting that the frequency of haze events with a significant amount of BC as the primary pollutant in Xuzhou in winter is comparatively high, consistent with aerosols from the source regions found below.

Model simulation results showed that the upper aerosol layers floating above the PBL were attached to the surface and that these layers interacted with each other up to a height of 2500 m. When these layers mixed in the atmospheric boundary layer, their effect was to sharply increase the $PM_{2.5}$ concentration near the ground and lead to secondary inorganic production of nitrate and sulfate, which have detrimental effects on air quality. Consistently, during haze pollution episodes, the BC concentration kept increasing, as well as possibly coating due to the inorganic growth, both causing the gradual rise of AAOD.

The School of Environment Science and Spatial Informatics, CUMT (34.22°N, 117.14°E), was set as the ending point for 72 h backward trajectory clustering and WPSCF analysis of multiple pollution events that were observed in this work. The results reveal that the wind sources differed in scenarios with and without particles aloft. From 2 to 10 December 2014, the air parcels aloft originating from Inner Mongolia and Mongolia accounted for the largest proportion, reaching 38% of the total, while from 1 to 10 January 2015, the air parcels aloft originating from the Xinjiang Uygur Autonomous Region accounted for the largest proportion, reaching 57%. Overall, the WPSCF analysis reveals that Shanxi, Hebei, Shandong, and Anhui are the most probable sources of $PM_{2.5}$ during these haze pollution episodes, which have emission profiles consistent with the properties of AOD, AAOD, AE, SSA, and FMF observed.

**Supplementary Materials:** The following supporting information can be downloaded at: https://www.mdpi.com/article/10.3390/rs14071589/s1, Figure S1: Volume size distribution from 0.05 to 15 μm (dV/dln(r) ($\mu m^3/\mu m^2$)) during 2–10 December 2014 and 1–10 January 2015 (China Standard Time (CST)); Figure S2: Comparison between model simulation and China Environmental Monitoring Station observation value of $PM_{2.5}$ concentration. References [36,67,68] are cited in the supplementary materials.

**Author Contributions:** Conceptualization, K.Q. and S.L.; methodology, K.Q., Q.H. and Y.Z.; investigation, Q.H. and Y.Z.; writing—original draft preparation, K.Q., Q.H. and Y.Z. writing—review and editing, S.L., P.T. and J.B.C.; visualization, K.Q., Q.H. and Y.Z.; supervision, J.B.C. All authors have read and agreed to the published version of the manuscript.

**Funding:** This research was funded by the National Natural Science Foundation of China, grant number 41975041.

**Institutional Review Board Statement:** Not applicable.

**Informed Consent Statement:** Not applicable.

**Data Availability Statement:** We thank the GSFC/NASA AERONET group for processing the AERONET data (http://aeronet.gsfc.nasa.gov, last accessed on 20 March 2022) and to the CALIPSO (https://eosweb.larc.nasa.gov/, last accessed on 20 March 2022) science team for the provision of publicly available data set.

**Acknowledgments:** We thank the GSFC/NASA AERONET group for processing the AERONET data (http://aeronet.gsfc.nasa.gov, last accessed on 20 March 2022) and to the CALIPSO (https://eosweb.larc.nasa.gov/, last accessed on 20 March 2022) for the provision of publicly available data set.

**Conflicts of Interest:** The authors declare no conflict of interest.

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
