# Peer review of "Aloft Transport of Haze Aerosols to Xuzhou, Eastern China: Optical Properties, Sources, Type, and Components"

_remotesensing, doi:10.3390/rs14071589_

Round 1
Reviewer 1 Report
Dear the Authors,
I think the English written of this manuscript is fine and I recommend this manuscript is accepted and published in the journal of Remote Sensing after some minor revisions.
Best Wishes,

Author Response
Dear Reviewer,
Thank you very much for your valuable comment on our manuscript. We carefully considered the feedback and made changes that we hope will be accepted. All updates make use of Microsoft Word's "Track Changes" feature to make changes apparent. This file contains a response-to-reviewer that includes a point-by-point answer to the remarks.
Comment 1. Page 4, 2.1 Ground-based observations, Lines 134-136: ‘Therefore, the solution is given in terms of the so-called Lidar Ratio (LR), which expresses the ratio between the extinction and backscatter coefficient, which must be assumed along with the vertical profile.’ ---This sentence should be revised to something more reasonable.
Response: Thank you for pointing out, we have revised our sentences as per your suggestion, in the manuscript. The revised sentences are as follows:
“The methodology shows large uncertainties because it requires solving a transcendent equation with two unknowns (extinction coefficient and backscattering coefficient). In order to solve this, a conjecture, Lidar ratio (LR) value, which is the ratio between the extinction and backscatter coefficient, along the vertical profile must be assumed.”
Comment 2. Page 15, Conclusion, Line 458: ‘indorganic’ should be ‘inorganic’?
Response: The typo error has been corrected in the revised manuscript.
Comment 3. Page 15, Lines 464-466: ‘The results show that the wind sources were different between cases where there were particles aloft and there were not particles aloft’ Change to ‘The results show that the wind sources were different between cases with and without particles aloft.’ Or revise to something more reasonable.
Response: We have revised our sentences as per your suggestion, in the manuscript. The revised sentence is as follows:
“The results reveal that the wind sources differed in scenarios with and without particles aloft.”
Comment 4. Page 15, Conclusion, Line 471: change ‘whch’ to ‘which’
Response: The typo error has been corrected in the revised manuscript.
Reviewer 2 Report
In this study, the authors perform a synergetic approach to investigate aerosol aloft transport events over Xuzhou (China). The optical properties, sources and types are studied in detail by using several instruments, networks and satellite measurements as lidar, AERONET, WRF-Chem, CALIPSO and HYSPLITT. I would only suggest making a small transition from the more general comments to the more specific examples in the Introduction, but this is optional. In general, I think that the study is complete, the paper is very well written and easy to read. I believe this paper is of great interest to the journal and I recommend to accept it in the present form.
Author Response
Thank you very much for the positive feedback. In the new version, we added a transition from the general comments to the examples
Reviewer 3 Report
This paper well describes a study on aerosol aloft transport using a variety of observation and simulation approaches which enlist surface measurements in Xuzhou (eastern China), satellite observations (MODIS and CALIPSO), and WRF-chem simulations for two events in 2014 and 2015 winter seasons. I'd like to suggest a few minor revisions for its publication as below.
Figure S2: I cannot find supplementary materials in this manuscript package somehow. Please add Figure S2 (Line 331) as one of main figures, not a supplementary material together with CALIPSO tracks which will be critical to figure out the big pictures of both events in 2014 and 2015 and helpful to better understand CALIPSO data shown in Figures 5 - 7.
Minor corrections:
Line 45: Remove the first “MPL”
Citation formats (such as Line 51 and Line 330): All the “et al,” will be reformatted in the final edition as [1], [2], right?
Line 54: Add "(Cloud-Aerosol Lidar Infrared Pathfinder Satellite) " which is currently shown in 2.2 later. Add a reference here or address something like that this will be described in 2.2 for more details.
Line 56: Add "(particulate matter)" to PM because it appears here first. Is this PM2.5?
Line 64: Remove “]”
Line 75: "particulate matter with diameters smaller than 2.5 μm in aero-75 dynamics) " -> See line 56.
Line 92: "a large number of local emissions sources" such as?"
Line 101: “study” rather than "manuscript"
line 113: MPL was already defined at line 45.
Line 244: Define AAOD here.
Line 245 Angstrm exponent -> AE (already defined)
Line 267: Add "(China Standard Time; CST)". Please check and be consistent to use the time unit (for example see Line 271 and Line 289).
Line 327: Add a reference for MODIS and the satellite name (probably Aqua or Terra and Aqua if both are used).
Line 339: Remove “.”
Line 340: 2nd and 4th
Figure 5 & 6: Missing labels for x-axes.
Figure 7 needs to have higher resolutions and bigger fonts for the labels.
Line 395: Define WPSCF
Line 396: It will be good to include more descriptions for four clusters seen in this study (Fig 8) including regions and general characteristics, which will be greatly helpful for readers to understand the trajectory results in Section 3.3.
Author Response
Responses to Reviewer 3.
Dear Reviewer,
Thank you very much for your valuable comment on our manuscript. We carefully considered the feedback and made changes accordingly. This file contains a response-to-reviewer that includes a point-by-point answer to the remarks.
Comment 1. Figure S2: I cannot find supplementary materials in this manuscript package somehow. Please add Figure S2 (Line 331) as one of main figures, not a supplementary material together with CALIPSO tracks which will be critical to figure out the big pictures of both events in 2014 and 2015 and helpful to better understand CALIPSO data shown in Figures 5 - 7.
The figure is are now added to the main text to enhance manuscript readiness. CALIPSO Tracks are shown in Figure 8 (former Figure 7). The figure is also enhanced for a better view.
Minor Correction:
Line 45: Remove the first “MPL”
Changed accordin
Comment 2. Citation formats (such as Line 51 and Line 330): All the “et al,” will be reformatted in the final edition as [1], [2], right?
Response: Yes, In the final version of the paper, if accepted, citation format will be revised to the specific format of the journal.
Response: Modification as per suggestion has been conducted in the revised manuscript. The revision of Line 54 is:
“Kar et al., (2015) detected strong pollutant outflows from the Mexico City metropolitan area during winter using Cloud-Aerosol Lidar Infrared Pathfinder Satellite (CALIPSO) lidar measurements (details of CALIPSO are described in 2.2), which often flowed as far north as the Texas coast
Comment 4. Line 56: Add "(particulate matter)" to PM because it appears here first. Is this PM2.5?
Response: The elaboration is added along with the abbreviation. The AOD-PM signifies for both AOD-PM2.5 and AOD-PM10.
“Han et al., (2015), studied and suggested that the correlations for AOD–PM10 and AOD–PM2.5 can be much improved if the aloft-aerosol layer is screened away”.
Comment 5. Line 75: "particulate matter with diameters smaller than 2.5 μm in aero-75 dynamics) " -> See line 56.
Response: line 56 has been rephrased with respect to the PM acronym and the response to Comment 4 highlights the AOD-PM term. We wish to provide a simple definition of PM2.5 in this line.
Comment 6. Line 92: "a large number of local emissions sources" such as?"
Response: We have highlighted the major local emission sources in Xuzhou in the revised version of the manuscript.
The sentence has now been revised as follows:
“Xuzhou, a heavily industrialized metropolitan area, hosts a large number of local emissions sources accounted mostly by the coal mining and combustion, construction projects for coal-fired power stations and industrial smoke (Chen et al., 2019; 2020a; 2020b; Liu & Gao, 2018; Qin et al., 2018).”
Comment 7. Line 101: “study” rather than "manuscript"
Response: Thank you for pointing out, we have made the modification in the revised version of the manuscript.
Comment 8. line 113: MPL was already defined at line 45.
Response: Thank you for pointing out, we have made the modification in the revised version of the manuscript.
Comment 9. Line 244: Define AAOD here.
Response: Thank you for pointing out, we have made the modification in the revised version of the manuscript.
Comment 10. Line 244: Define AAOD here.
Response: Thank you for pointing out, we have made the modification in the revised version of the manuscript.
Comment 11. Line 245 Angstrom exponent -> AE (already defined)
Response: Thank you for pointing out, we have made the modification in the revised version of the manuscript. We have abbreviated Angstrom exponent to AE in the abstract and have used AE thereafter.
Comment 12. Line 267: Add "(China Standard Time; CST)". Please check and be consistent to use the time unit (for example see Line 271 and Line 289).
Response: Modification as per suggestion has been incorporated in the revised manuscript.
Comment 13. Line 327: Add a reference for MODIS and the satellite name (probably Aqua or Terra and Aqua if both are used).
Response: The MODIS AQUA true-color images were used. It has been mentioned in the supplementary section of the paper. The citation as suggested has been included in the main manuscript.
Comment 13. Line 339: Remove “.”
Response: Thank you for pointing it out. It has been corrected in the revised version of the manuscript.
Comment 14. Line 340: 2nd and 4th
Response: Thank you for pointing it out. It has been corrected in the revised version of the manuscript.
Comment 15. Figure 5 & 6: Missing labels for x-axes.
Comment 16. Define WPSCF.
Response: Thank you for pointing it out. It has been the elaboration of the abbreviation is added in the revised version of the manuscript.
Comment 17. Line 396: It will be good to include more descriptions for four clusters seen in this study (Fig 8) including regions and general characteristics, which will be helpful for readers to understand the trajectory results in Section 3.3.
Response: Thank you for the suggestion regarding the discussion of trajectory clusters and their geographic characteristics. We have incorporated this in the Discussion section of the paper.